# Diversity Analysis and Function Prediction of Bacterial Communities in the Different Colored Pericarp of *Citrus reticulata* cv. ‘Shatangju’ Due to ‘*Candidatus* Liberibacter asiaticus’ Infection

**DOI:** 10.3390/ijms241411472

**Published:** 2023-07-14

**Authors:** Feiyan Wang, Congyi Zhu, Ruimin Zhang, Yongjing Huang, Wen Wu, Jiezhong Chen, Jiwu Zeng

**Affiliations:** 1Key Laboratory of South Subtropical Fruit Biology and Genetic Resource Utilization & Guangdong Province Key Laboratory of Tropical and Subtropical Fruit Tree Research, Institute of Fruit Tree Research, Guangdong Academy of Agricultural Sciences, Guangzhou 510640, China; wfei08@163.com (F.W.); zhucongyi@hotmail.com (C.Z.); xmingz@163.com (R.Z.); yjhgzh@126.com (Y.H.); wuwen8@126.com (W.W.); 2Guangdong Biomaterials Engineering Technology Research Center, Institute of Biological and Medical Engineering, Guangdong Academy of Sciences, Guangzhou 510316, China; 3College of Horticulture, South China Agricultural University, Guangzhou 510642, China; cjzlxb@scau.edu.cn

**Keywords:** citrus, pericarp, microbiome, pigment, Huanglongbing

## Abstract

Huanglongbing (HLB), caused by the *Candidatus* Liberibacter spp., is the most devastating disease in the citrus industry. HLB significantly affects and alters the microbial community structure or potential function of the microbial community of leaves and roots. However, it is unknown how the microbial community structure of the pericarp with different pigments is affected by *Candidatus* Liberibacter asiaticus (*C*Las). This study identified the enriched taxa of the microbial community in the citrus pericarp with normal or abnormal pigment and determine the effects of HLB on the pericarp microbial community using 16S rRNA-seq. The alpha and beta diversity and composition of microbial communities were significantly different between normal and abnormal pigment pericarp tissues of ripe fruits infected by *C*Las. *Firmicutes*, *Actinobacteriota*, *Bacteroidota*, *Acidobacteriota*, and *Desulfobacterota* dominated the pericarp microbiota composition in WDYFs (whole dark yellow fruits) samples. The relative abundance of most genera in WDYFs was higher than 1%, such as *Burkholderia*, and *Pelomonas*. However, with the exception of the HLB pathogen, the relative abundance of most genera in the abnormal-colored pericarp samples was less than 1%. *C*Las decreased the relative abundance of pericarp taxonomic. The predicted function of microbial was more plentiful and functional properties in the WDYF sample, such as translation, ribosomal structure and biogenesis, amino acid transport and metabolism, energy production and conversion, and some other clusters of orthologous groups (COG) except for cell motility. The results of this study offer novel insights into understanding the composition of microbial communities of the *C*Las-affected citrus pericarps and contribute to the development of biological control strategies for citrus against Huanglongbing.

## 1. Introduction

Microorganisms form microbial communities of plant, animal, or ecological environments, which are collectively known as microbiomes. Microorganisms can be associated with plants in different compartments, including the rhizosphere, rhizoplane, phyllosphere, and endosphere [1,2]. The plant microbiomes can perform a variety of functions on plants, such as supplying nutrition for plants, improving tolerance against biological and abiotic stresses through phytohormones, and producing antimicrobial compounds or stimulating plant immunity to improve disease resistance [3,4,5]. Therefore, the plant microbiome plays an important role in the plant life cycle, also considered to be an integral component of the host plant. Understanding the plant microbiome is of great significance for exploring its functions for plants.

The microbiome of different citrus tissues shows high diversity and performs a variety of functions, such as in leaves and roots [6,7,8]. The strains *Variovorax*, *Methylobacillus*, *Novosphingobium*, and *Methylotenera* are involved in promoting plant growth and antibiotic production [8]. The rhizosphere strain *Burkholderia* can trigger the expression patterns of genes involved in systemic resistance in inoculated plants [9]. HLB, caused by the *Candidatus* Liberibacter spp., is the most devastating disease in the citrus industry worldwide [10]. HLB can impair the phloem transportation of photoassimilates [10], cause root decline [11], and decrease the quality and yield of fruit [12]. To date, a plethora of studies on the microbial community of different tissues associated with HLB have found that HLB can significantly affect the microbial community structure or functional property [6,7,8]. Some beneficial microbes, such as *Bacillus* sp., can induce host defense responses against *Candidatus* Liberibacter asiaticus (*C*Las), or the benign *Xylella fastidiosa* strain EB92-1 can reduce the incidence of HLB symptoms for several years in mature trees and newly planted young trees [13,14,15]. *Bacillus* sp. can induce host defense responses against *C*Las by enhancing the expression of several transcription factors and enriching some metabolites involved in disease resistance [13,14]. These studies show that the microbial community in different tissues of citrus has a significant relationship with the existence of HLB and the pathogen.

The previous studies about the microbial community structure of citrus mainly focus on the endosphere in leaves or rhizosphere in roots [9,15,16]. The microbial community of different color pericarps associated with *C*Las has not been reported. The present works have studied the effect of *C*Las on bacterial diversity using 16S ribosomal DNA (rDNA)–based microbiota analysis, and explore the predicted function of core microorganisms in the citrus pericarps infected by *C*Las. These results indicate that *C*Las can not only affect the pigmentation of the fruit exocarp (the outer peel) but can also affect the diversity of bacteria in the entire citrus fruit pericarp.

## 2. Results

### 2.1. Difference in the Microbial Diversity of the Citrus Pericarp with Different Pigment

A total of 1,158,794 raw reads of 16S rRNA was obtained from sixteen pericarp samples (four repetitions of each pericarp group) with an average of 72,425 reads of the bacteria per sample. The average read length of 16S rRNA was 387 bp. The amplicon sequence variants (ASV) table containing 5848 ASVs (Appendix A) was obtained. A total of 64 ASVs coexisted in all pericarp samples (Figure 1B). The most abnormally colored pericarps, ‘whole green fruits (WGFs)’, and the normal pericarps, ‘whole dark yellow fruits (WDYFs), presented many unique ASVs compared to the other groups (WGF:1801, WDYF:2317, Figure 1B). The annotated ASVs were analyzed by Venn, which showed 202 genera overlapped in all samples, and each group of samples has its unique genera (Figure 1C). Combined with the pericarp phenotype, the genera may contribute to the color phenotype. This is a question that deserves further study.

The Simpson Index and Shannon Index measure microbial alpha diversity. The Simpson and Shannon indices revealed significant differences in the overall bacterial community of citrus fruit pericarp samples with different pigments at the ASV level (Figure 2A,B). However, TYBGFs vs. WLYFs did not present significant differences between each other (Simpson Index *p*-value = 0.43, Shannon Index *p*-value = 0.54). Therefore, the alpha diversity of WDYFs was remarkably different from the diversity of other samples (Figure 2, Appendix A).

Principal coordinate analysis (PCoA) and partial least squares discriminant analysis (PLS-DA) of the bacterial communities revealed the segregation and clustering of samples representing the different colored pericarps (Figure 3). The PERMNOVA analysis showed a significant effect of the different colored pericarps on the beta diversity of bacterial communities (R^2^ = 0.7185, *p*-value = 0.001). In addition, the hierarchical clustering of bacteria shows better separations between the normal (WDYFs) and the abnormal color pericarps (Appendix A).

### 2.2. Differences in the Relative Abundance of Taxa in the Microbiome of Different Colored Pericarps

The bacterial ASVs were assigned to 40 phyla and 962 genera in all pericarp samples. Circos plots showed that the abnormal color pericarp samples (WGFs, TYBGFs, and WLYFs) presented a higher association with *Proteobacteria* (approximately 25% in WGFs, 31% in TYBGFs, and 30% in WLYFs) and *Proteobacteria* was only 14% in WDYFs at the phylum level. The association with *Firmicutes*, *Actinobacteriota*, and *Bacteroidota* was approximately 29%, 17%, and 38% in WGFs, respectively, while they were less than 10% in TYBGFs and WLYFs (Appendix A). *Acidobacteriota* and *Desulfobacterota* were less than 10% in all abnormal color pericarp samples (Appendix A). *Firmicutes*, *Actinobacteriota*, *Bacteroidota*, *Acidobacteriota*, and *Desulfobacterota* dominated the pericarp microbiota compositions at the phylum level in WDYFs samples, contributing to 61%, 77%, 57%, 92%, and 94%, respectively (Appendix A).

At the genus level of bacteria, 202 genera coexisted in the four types of pericarp samples. They also have their genera (WGFs: 189, TYBGFs: 25, WLYFs: 40, and WDYFs: 165) in all pericarp samples (Figure 1C). The relative abundance of some bacterial genera was obviously different among the four types of pericarps (Appendix A). *Candidatus* Liberibacter spp. was the most abundant genus (WGFs: 72.26%, TYBGFs: 91.25%, WLYFs: 95.57%) in all abnormal fruit pericarps and was only 0.22% in WDYFs (Appendix A). Except for *Sphingomonas* (4.02%) in TYBGFs, the relative abundance of most genera in the TYBGFs, and WLYFs samples was less than 1%. The bacterial composition of the WGFs samples was more complex than that in the TYBGFs and WLYFs. Some genera that the relative abundance was more than 1% in WGFs samples included *Bacteroides* (3.46%), *Pseudomonas* (2.45%), *Prevotella* (1.27%), *Geobacillus* (1.31%), *Delftia* (1.00%) (Appendix A). On the contrary, the relative abundance of most genera in WDYFs is higher than 1%, such as *unclassified_f_Rhodocyclaceae* (6.44%), *Staphylococcus (*4.03%), *Thauera* (2.85%), *Zoogloea* (2.91%), *Raoultella* (1.54%), *Paludibacter* (2.31%), *Burkholderia-Caballeronia-Paraburkholderia* (2.37%), and *Acidothermus (*2.41%, Appendix A). Some genera that had a very low percentage (less than 0.01%) were merged into others (WGFs: 15.42%, TYBGFs: 3.12%, WLYFs: 2.74%, and WDYFs: 48.86%, Appendix A). Therefore, the microbial structure of the WDYFs pericarp was more complex at the genus level. Although the relative abundance of *Candidatus* Liberibacter spp. is inconsistent with color severity based on the result data, *Candidatus* Liberibacter spp. should have a significant effect on fruit phenotype.

Some genera showed significant differences among all samples. *Candidatus* Liberibacter spp. is the pathogen of Huanglongbing. It reveals significant differences in the relative abundance among the four types of pericarp samples (*p* = 0.0072) and was significantly low in abundance in the WDYFs samples (Appendix A). Some other genera also show significant differences. The genera *unclassified_Rhodocyclaceae*, *Thauera*, *Zoogloea*, and *Paludibacter* were significantly high in abundance in WDYFs. *Prevotella*, *Bacteroides*, and *Pseudomonas* showed the same significantly high abundance in the WGFs samples (Appendix A). TYBGFs samples possess a significantly enriched genus, *Sphingomonas*. The *p*-values of all the above genera were less than 0.05. The differences in pericarp microbiota among all pericarp samples were also evidenced by LEfSe (linear discriminant analysis effect size), which showed the most differently abundant taxa of these pericarp samples (Appendix A). Using a metagenomic biomarker discovery approach, some genera were discovered to be significantly enriched in the WDYFs pericarps (linear discriminant analysis (LDA) score > 4, *p*-value < 0.05), such as *Rhodocyclaceae* (LDA score = 4.44, *p*-value = 0.0048), *Enterobacteriaceae* (LDA score = 4.17, *p*-value = 0.0051), *Zoogloea* (LDA score = 4.06, *p*-value = 0.0065), *Thauera* (LDA score = 4.00, *p*-value = 0.0072), and *Comamonadaceae* (LDA score = 4.25, *p*-value = 0.0084). The WGFs pericarps also own some significantly enriched bacterial genera, including *Bacteroidaceae* (LDA score = 4.23, *p*-value = 0.039) and *Pseudomonadales* (LDA score = 4.10, *p*-value = 0.0083) at the genus level (Appendix A). These genera may be involved in the formation of fruit color phenotype and were worthy of further study.

### 2.3. Correlation Analysis between Microbes and Metabolites of Citrus Pericarps

The correlation analysis between the pericarps’ microbes and metabolites indicates potential mutual contribution. To inquire into the contributions of metabolites to pericarps’ microbes, RDA was carried out on the top significantly changed metabolites (docosanoic acid, quinine, dihydropinosylvin, coronatine, peonidin-o-hexoside, 6,7-dimethoxy-4-methylcoumarin&quot, myricetin, and 3-indole propionic acid) related to the top pericarps’ microbes. The top high-enriched or low-enriched metabolites that have been published before [17] significantly correlated to the pericarps’ microbes, especially 3-indolepropionic acid (R^2^ = 0.9873, *p*-value = 0.001), quinine (R^2^ = 0.8621, *p*-value = 0.002), and myricetin (R^2^ = 0.9872, *p*-value = 0.002). This study found that the top three high-enrichment metabolites in WDYFs, namely 3-indole propionic acid, 6,7-dimethoxy-4-methylcoumarin, and myricetin significantly correlated to the bacteria of the WDYFs pericarp. However, the top five high-enrichment metabolites in the three abnormal pericarps, namely docosanol acid, dihydropinosylvin, coronatine, quinine, and peonidin-o-hexoside significantly correlated to bacteria of WGFs, TYBGFs, and WLYFs samples (Figure 4).

To investigate how the pericarps’ microbes are related to different patterns of metabolites due to ‘*Candidatus* Liberibacter asiaticus’ infection in ‘Shatangju’ mandarin fruits, the correlation between the top 50 genera and the top 50 significantly changed metabolites that have been published before [17] was analyzed using the Spearman rank correlation coefficient (Appendix A). In Appendix A, the relative abundance of some genera, such as *Faecalibacterium*, *Curtobacterium*, *Staphylococcus*, *Raoultella*, *unclassified_f_Prevotellaceae*, *Bacillus*, *Chryseobacterium*, *Brevundimona*, and *Candidatus* Liberibacter, etc. were significantly correlated with the concentrations of the some significantly changed metabolites enriched in the four types of pericarps, such as o-caffeoyl maltotriose, tricin 4’-o-β-guaiacylglycerol, n-hexosyl-p*-*coumaroyl serotonin, morroniside, 6,7-dimethoxy-4-methylcoumarin, and p*-*coumaraldehyde. Interestingly, this study found that *Faecalibacterium* was positively or negatively correlated with all metabolites except for d-pantothenic acid, o-caffeoyl maltotriose, and tricin 4’-o-β-guaiacylglycerol. There were four significantly changed metabolites including d-pantothenic acid, quinine, myricetin, and 3-indole propionic acid that were significantly correlated with the relative abundance of most bacterial genera, such as *Staphylococcus*. Myricetin and 3-indole propionic acid were significantly positively correlated with the relative abundance of most bacterial genera except *Faecalibacterium*, *Candidatus* Liberibacter, and *norank_f_Mitochondria* that were a negative correlation with the two metabolites (Appendix A).

### 2.4. Bacterial Function Prediction in the Different Colored Citrus Pericarps

Functional annotation was performed by comparing clean reads to the COG (clusters of orthologous groups) databases. Twelve abundant COG functions (relative abundance ≥ 5%) were identified in all pericarp samples (Appendix A). The most abundant was the translation, ribosomal structure, and biogenesis (J) item (WDYF: 6.94%; WLYF: 14.1%; TYBGF: 12.76%; WGF: 10.22%), except for the function unknown (S) item, followed by amino acid transport and metabolism (E), energy production and conversion (C), etc. (Appendix A). The results of the COG functional annotation showed that the pericarp groups had functional differences in the field of 23 COG functional annotations (Appendix A). The orders of the relative proportion of COG functional annotations (from high to low) differed among the four group samples (Figure 5). The result revealed that the COGs in the WDYFs pericarp were always significantly different from the other pericarp samples except for cell motility (N).

Six level_1-pathways were identified through KEGG database analysis (Figure 6A). The annotated unigenes of the four types of citrus pericarps were mainly involved in the metabolic pathway, and the proportions of the metabolic pathway in WDYFs was 76.25%; in WLYFs was 73.81%; in TYBGFs was 74.03%; and in WGFs was 75.78%, followed by environmental information processing, genetic information processing, cellular processes (Figure 6). Except for genetic information processing, the proportions of other level_1-pathways in WDYFs were significantly higher than that in WLYFs, TYBGFs, and WGFs (*p*-value < 0.05). The functional annotation of microorganisms is further analyzed in KEGG level_2-pathways. There are 46 KEGG level-2 pathways that were annotated in all pericarp samples (Figure 6B). They are mainly involved in carbohydrate metabolism, amino acid metabolism, energy metabolism, and the metabolism of cofactors and vitamins.

There are five level_3-pathways that were significantly different among the four groups of pericarps based on the KEGG database (*p*-value < 0.05) (Figure 7). Among them, ko01120 (microbial metabolism in diverse environments) and ko01230 (biosynthesis of amino acids) in WDYFs were significantly higher than that in TYBGFs and WLYFs and others were significantly lower than that in the TYBGFs and WLYFs groups (*p*-value < 0.05). Thirteen enzymes differed greatly among the four groups of pericarps (*p*-value < 0.05) (Figure 8). The histidine kinase and peptidylprolyl isomerase in the WDYF were significantly higher than that in the other three groups (*p*-value < 0.05) (Figure 8). Eleven enzymes in TYBGFs, WGFs, and WLYs were significantly higher than that in WDYFs (*p-*value < 0.05) (Figure 8C,D). Ten KEGG ortholog (KO) groups were significantly different among the samples (*p*-value < 0.05) (Figure 8A,B). Three KOs in the WDYFs were higher than those in the other three groups. Their functions were related to RNA polymerase sigma-70 factor, ECF subfamily (K03088), ABC-2 type transport system ATP-binding protein (K01990), and ABC-2 type transport system permease protein (K01992). Seven KOs in TYBGFs, WGFs, and WLYFs groups were significantly different from that in the WDYFs group that reveal a high abundance trend (*p*-value < 0.05).

## 3. Discussion

Huanglongbing (HLB) can greatly affect the leaves, roots, and fruits. Previous studies focused on the microbial community of the citrus endophyte of leaves and root or rhizosphere associated with HLB and *C*Las suppression, and *C*Las significantly affected the microbial community structure or function [6,7,8]. The current study carried out a comprehensive analysis of the pericarp microbiome of ‘Shatangju’ mandarin with different pigments infected by *C*Las. The resulting data showed significant differences in the microbial alpha and beta diversity and composition of microbial community between normal and abnormal pigment pericarp tissues of ripe fruits infected by *C*Las. The significant overall differences based on the Simpson Index and Shannon Index among the different colored pericarp samples were observed, except TYBGFs vs. WLYFs. These results highlight the great correlation between bacterial communities and the pericarp phenotype. In rhizosphere and rhizoplane samples, microbiological compositions were dramatically influenced by *C*Las using a 16S rDNA-based analysis [18]. The current results are consistent with previous reports. However, the significant difference in alpha diversity of the citrus rhizosphere and rhizoplane microbiome due to HLB was not observed using metagenomic (MG) and metatranscriptomic (MT) approaches [9]. The difference in the impact of HLB on the rhizosphere microbiome might be due to the genomic background, tree age, and the development stage of *Candidatus* Liberibacter spp. infection [18,19]. In this study, the bacterial microbiome of the ‘Shatangju’ mandarin pericarp with different pigments was affected by *C*Las and showed a great correlation with the pericarp phenotype using 16S rRNA sequencing. Combined with the previous studies, more accurate techniques, such as metagenomic, may be needed to further analyze the relationship between Huanglongbing and the pericarp microbiome. Regarding the bacterial community of citrus pericarp, *Proteobacteria*, *Actinobacteria*, *Bacteroidetes*, and *Firmicutes* were the predominant genera that have been reported on the fruit surface of oranges [20]. In this study, *Proteobacteria* was the predominant genera in the WGFs and TYBGFs sample. *Firmicutes*, *Actinobacteriota*, *Bacteroidota*, *Acidobacteriota*, and *Desulfobacterota* dominated the pericarp microbiota compositions at the phylum level in the WDYFs samples. Therefore, *C*Las affected the main microbiota structure in ‘Shatangju’ mandarin pericarp tissues and reduced the diversity of microorganisms in the peel. The relationship between these genera and fruit phenotypes should deserve further study.

The plant microbiome plays an important role in plant health and defense against plant pathogens. Plant-associated microbiota can defend against plant pathogens via direct competition, producing antimicrobial compounds or stimulating plant immunity to resist or tolerate pathogen infection [3,4,5,21]. To date, a plethora of research topics involved in the identification of microbial members in citrus associated with *C*Las. However, *C*Las can seriously affect the microbial community and reduce the beneficial microorganism of citrus leaves or roots [6,7,8]. The beneficial bacteria genera were reduced by HLB and *C*Las include *Variovorax*, *Novosphingobium*, *Methylobacillus*, *Methylotenera*, *Burkholderia*, *Bacillus*, and *Lysobacters* [8,18]. They are involved in the competition for nutrition with pathogens, antagonizing pathogens through antibiotic production, assisting the host with nutrient acquisition, and inducing host defense responses [9,14]. These studies show that beneficial bacteria can be used to defend against *C*Las and improve plant health. In this study, the predominant genera in WDYF include *unclassified_Rhodocyclaceae*, *Staphylococcus*, *Thauera*, *Zoogloea*, *Raoultella*, *Paludibacter*, *Burkholderia-Caballeronia-Paraburkholderia*, *Pelomonas*, and *Acidothermus*. According to previous research reports, the endophytes, such as *Methylpbacterium*, *Burkholderia*, *Sphingomonas*, and *Bradyrhizobiaceae* were involved in nutrition competition, antagonizing pathogens, and host disease-resistant response [22]. *Burkholderia* can trigger the expression of defense-related genes and SA-mediated induced system resistance genes [9]. The predominant *Burkholderia* genus, such as *Burkholderia-Caballeronia-Paraburkholderia*, may play a positive role in the health of citrus fruit [23].

## 4. Materials and Methods

### 4.1. Sample Collection

*Citrus reticulata* cv. ‘Shatangju’ that was used for this study was infected by the HLB-pathogen, *Candidatus* Liberibacter asiaticus, and grown at the same citrus research orchard. At maturity, the *C*Las-infected trees produced different colored fruits and they were classified into four types based on the pericarp coloration: ‘whole green fruits (WGFs)’, ‘top yellow and base green fruits (TYBGFs)’, ‘whole light yellow fruits (WLYFs)’, and ‘whole dark yellow fruits (WDYFs, Figure 1A). The entire pericarps including albedo and flavedo tissue together were collected and used for this study. Detailed information on the geographical location and handling of the relevant materials refers to the published paper [23].

### 4.2. DNA Extraction and 16S rRNA Gene Amplification

Total microbial genomic DNA was extracted from the pericarp samples using the FastDNA^®^ Spin Kit (MP Biomedicals, Santa Ana, CA, USA) according to the manufacturer’s instructions. The hypervariable region V5–V7 of bacterial 16S rRNA gene was amplified with primer pairs 799F (5’-AACMGGATTAGATACCCKG-3’) and 1193R (5’-ACGTCATCCCCACCTTCC-3’) by using the TransStart^®^ FastPfu DNA Polymerase (TransGen Biotech, Beijing, China) according to the manufacturer’s protocols and ABI GeneAmp® 9700 PCR thermocycler (ABI, Foster City, CA, USA) [24].

### 4.3. Illumina MiSeq Sequencing and Data Processing

The purified amplicons were pooled in equimolar amounts and paired-end sequenced on an Illumina MiSeq PE3000 platform (Illumina, San Diego, CA, USA) according to the standard protocols by Majorbio Bio-Pharm Technology Co., Ltd. (Shanghai, China). After demultiplexing, the resulting sequences were quality filtered with fastp (0.19.6) [25] and merged with FLASH (v1.2.11) [26]. Then, the high-quality sequences were de-noised using the DADA2 plugin in the Qiime2 [27,28] pipeline with recommended parameters, which obtains single nucleotide resolution based on error profiles within samples. DADA2-denoised sequences are usually called amplicon sequence variants (ASVs). To minimize the effects of sequencing depth on alpha and beta diversity measure, the number of sequences from each sample was rarefied to 20,000, which still yielded an average Good’s coverage of 97.9%. The taxonomic assignment of ASVs was performed with the representative 99% similarity using the Naïve Bayes consensus taxonomy classifier and implemented in Qiime2 and the SILVA 16S rRNA database (v138). The metagenomic function was predicted by PICRUSt2 based on ASV representative sequences [29].

### 4.4. Statistical Analysis

The Majorbio Cloud platform (https://cloud.majorbio.com, accessed on 16 May 2021) was used to perform the bioinformatic analysis of the pericarp microbiota. Based on the ASVs information, the alpha-diversity indexes of Simpson and the Shannon index were calculated with Mothur v1.30.2 [30]. Principal coordinate analysis (PCoA) was used for judging the similarity among the four types of pericarp samples based on Bray–Curtis distances, and the PERMANOVA test was used to assess the percentage of variation using the Vegan v2.5-3 package through the adonis function. The linear discriminant analysis (LDA) effect size (LEfSe) was performed to search for the significantly different taxa (phylum to genera) of bacteria among the four types of pericarps (LDA score > 4, *p* < 0.05). The distance-based redundancy analysis (db-RDA) was performed using the Vegan v2.5-3 package to investigate the effect of pericarp metabolites on the bacterial community structure. The forward selection was based on Monte Carlo permutation tests (permutations = 9999). A correlation analysis between top-changed pericarp metabolites and top-enriched bacteria was performed through Spearman’s correlation coefficient with an *r* > 0.6 (*p* < 0.05).

## 5. Conclusions

In summary, this study demonstrates that the diversity and composition of microbial communities between normal and abnormal pigment pericarp tissues of ripe fruits infected by *C*Las revealed marked differences. The effect of *C*Las with different content on the relative abundance of the microbial was different. *Firmicutes*, *Actinobacteriota*, *Bacteroidota*, *Acidobacteriota*, and *Desulfobacterota* dominated the pericarp microbiota composition at the phylum level in WDYFs samples. The relative abundance of most genera in WDYFs is higher than 1%, such as *Burkholderia* and *Pelomonas*. The irregularly colored pericarp samples each have different dominant bacteria genera from WDYFs. The relationship between these genera and fruit phenotypes deserves further study. This study provides novel insights for understanding the composition of the *C*Las-affected citrus pericarps-enriched microbiome and its effect on plant health.

## Figures and Tables

**Figure 1 ijms-24-11472-f001:**
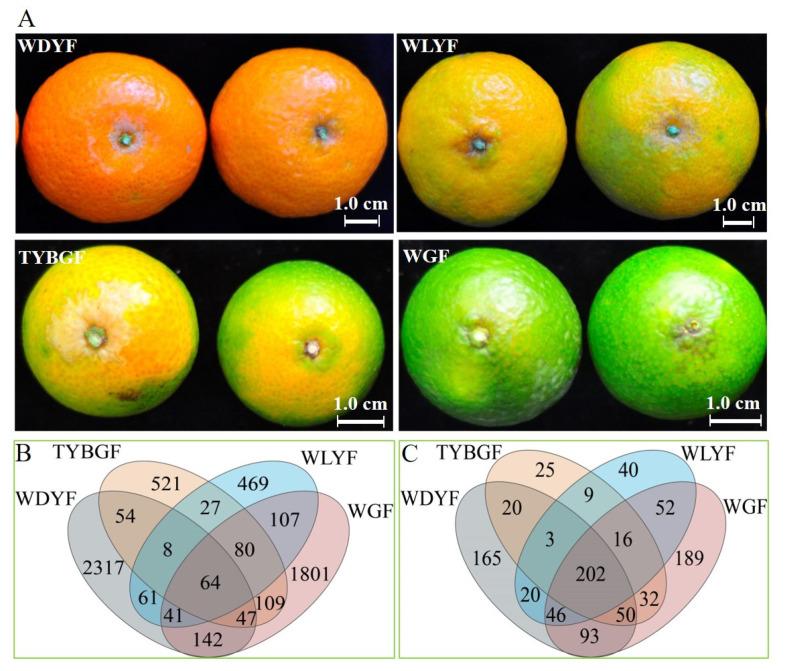
The phenotype of the four types of pericarps: ‘whole green fruits (WGFs)’, ‘top yellow and base green fruits (TYBGFs)’, ‘whole light yellow fruits (WLYFs)’, and ‘whole dark yellow fruits (WDYFs)’ (**A**), Venn plot of the bacterial ASVs (**B**), and Venn plot of the bacterial genus (**C**) in the WGFs, TYBGFs, WLYFs, and WDYFs pericarp.

**Figure 2 ijms-24-11472-f002:**
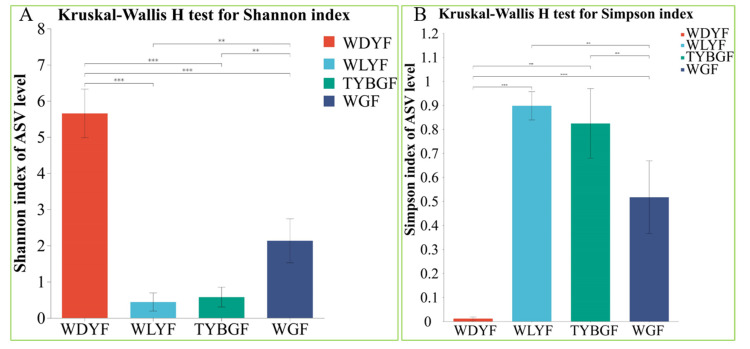
Histograms showing comparisons of alpha diversity of the bacterial community in ‘Shatangju’ mandarin fruit pericarp tissues with different pigments. (**A**) Shannon Index; (**B**) Simpson Index. Values indicate the *p*-value of the results of pairwise comparison using the Wilcoxon rank-sum test (**: adjust *p*-value < 0.01; ***: adjust *p*-value < 0.001).

**Figure 3 ijms-24-11472-f003:**
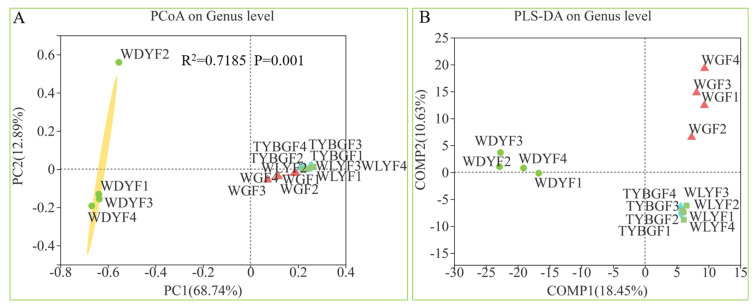
PCoA and PERMANOVA results (**A**), and PLS-DA (**B**) of the different colored pericarps (WGFs, TYBGFs, WLYFs, and WDYFs) on the bacterial communities.

**Figure 4 ijms-24-11472-f004:**
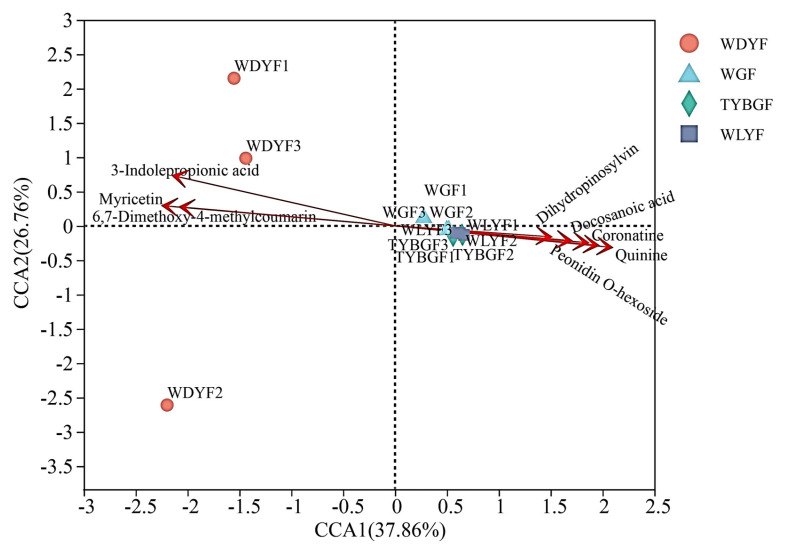
RDA plots relating the significantly changed metabolites of the pericarp and the relative abundance of bacteria that were enriched in pericarp samples.

**Figure 5 ijms-24-11472-f005:**
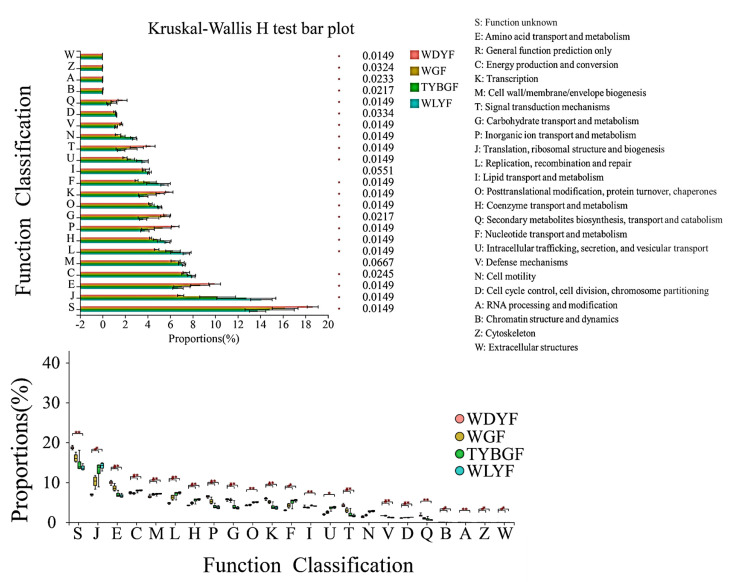
COGs function predicted for 16s rRNA of the four types of citrus pericarp by PICRUST. *: adjust *p-*value < 0.05; **: adjust *p-*value < 0.01.

**Figure 6 ijms-24-11472-f006:**
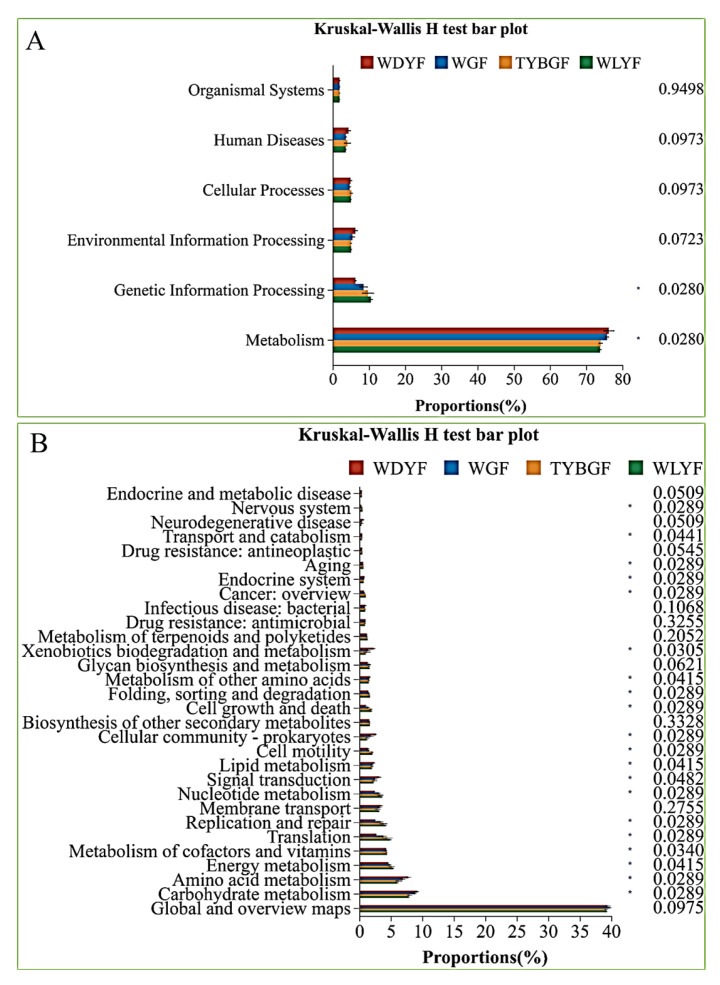
KEGG pathway level_1 (**A**) and level_2 (**B**) function predicted for 16s rRNA of the citrus pericarp by PICRUST. *: adjust *p-*value < 0.05.

**Figure 7 ijms-24-11472-f007:**
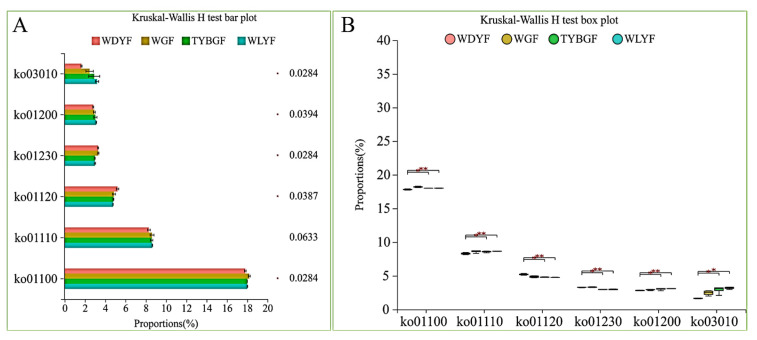
Significant difference in function predicted for KEGG pathway level_3. (**A**) histogram of function; (**B**) boxplot of function. *: adjust *p-*value < 0.05; **: adjust *p-*value < 0.01.

**Figure 8 ijms-24-11472-f008:**
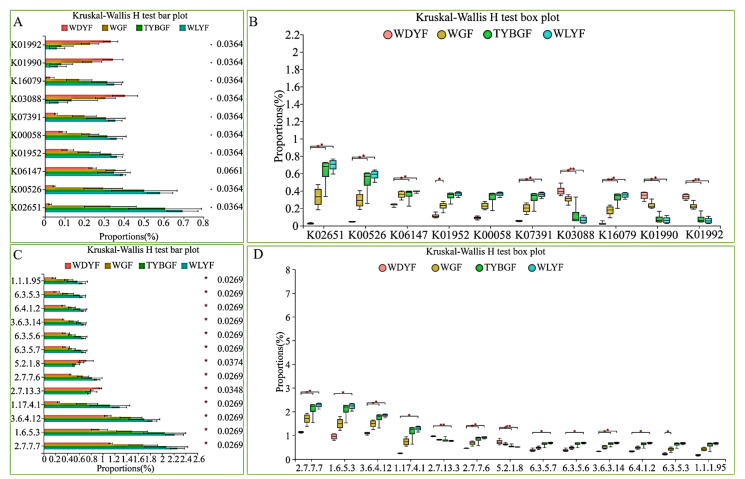
Significant difference in function predicted for KO (**A**,**B**), and enzyme (**C**,**D**). *: adjust *p-*value < 0.05; **: adjust *p-*value < 0.01.

## Data Availability

The datasets presented in this study can be found in online repositories. The names of the repository/repositories and accession number(s) can be found below: NCBI BioProject—PRJNA904534.

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
