# Peer review of "Diversity Analysis and Function Prediction of Bacterial Communities in the Different Colored Pericarp of Citrus reticulata cv. ‘Shatangju’ Due to ‘Candidatus Liberibacter asiaticus’ Infection"

_ijms, 2023, doi:10.3390/ijms241411472_

Round 1
Reviewer 1 Report
While the manuscript is well written, there is however a great deal of information is lacking
Line 63 in Intro and In Methods 4.1 Sample collection -Need to explain ‘pericarp’ sample:
What exactly did the authors Sample? The outer surface of the citrus fruit peel? The exocarp of citrus fruit is the outer colored peel (Sadka et al, 2019) often referred to as the flavedo (Schneider, 1968)
Or did they punch out tissue samples from the entire Pericarp? Through the peel? the definitions of the Pericarp, a diagram would be helpful as in (The citrus fruit, termed hesperidium, is a fleshy fruit which, like all berry-type fruit, is characterized by a thick and fleshy pericarp (Esau, 1966; Fahn, 1990). The pericarp is usually divided into three tissues: the exocarp, which is the outer skin, the mesocarp, which usually refers to the major fleshy, edible interior, and the endocarp, an internal tissue composed of one (as in tomato) or several cell layers. In true fruit, which develop from the ovary, these three tissues are part of the ovary wall. (Sadka et al, 2019 Front. Plant Sci. 10:1167.) Sadka A, Shlizerman L, Kamara I and Blumwald E (2019) Primary Metabolism in Citrus Fruit as Affected by Its Unique Structure. Front. Plant Sci. 10:1167. doi: 10.3389/fpls.2019.01167
Once concern for this study uses short-sequences (376bp) for 16s data analyses when current analyses using 16s requires >800bp or longer for bacterial classifications to ‘Genus’, and > 1,300 to full length for classification to ‘Specie’. This increased in the 1980s to >500bp for 16s data analyses, and again in early 2002 to longer sequences. Authors should use a more stringent statistical analyses values (BLAST matches should be < e-10, and not e-5, with the matches that are 99 to 100% identical sequence to provide a significant match to provide stronger support as a valid ‘Genus’ hit, and there should be at a minimum coverage at least 6 identical sequence hits to support that the identification is real.
Lines 137: reword sentence I think authors mean 'significantly lower in abundance in the WDYF.
Line 159 to 160- Author need correct reference from #23 to #22 or to one that supports the statement, and the sentence statement is too strong, as the research does not ‘show ‘contribution’, but the authors identified a ‘correlation’ between the top 8 (high and top 8 low metabolites? Reported in Ref#22: Wang et al, 2021 with the elevated microbes in the pericarp samples?).
Line 245 reference used #16 does not seem appropriate for statement.
Lines 255 -257: Cannot say HLB affected microbiome, but that ‘Fruit pericarp infected with CLas showed altered phytochemical profiles (ref#22 Wang et al), that correlated with reduce bacterial genre’.
Line 306-308: with such short sequence lengths ~376 bp, you need to have 99% or greater Identical match to make a call for bacterial Genus, Even so the results report weak 'hit' analyses, even though 97.9% looks good (standard for OTUs) it is not significant given their sequencing length coverage of the 16s Region. The analyses should use >500 bp (800 bp is standard for most Bacteria research) with short reads need to use 100% identical, contiguous sequence lengths for analyses.
Line 313-314. If the taxonomic assignment is weak -then all subsequent analyses is also weak, short sequence leads to poor analyses, and more software analyses using it-- continues to produce questionable conclusions. ReAnalyses using more stringent parameters are needed and then reanalyses of the ASV and other LDA score determinants for potential Genus matches. P values should be set for p<0.001.
Lines 320 and 325: with short sequences need to use the sequences with 100% identity and need to use a probability levels of <0.01 at a minimum. LDA and Spearman’s Correlation Coefficient.
Supplemental Figure S2. Needs a Color-Key to tell what the Colors are? to what Bacteria Phyla? Order? In each of the fruit pericarp sample groups.
When Reporting p-values are normally restricted to 3 or 4 decimal places, need to change throughout all figs and tables so as in Fig. S5, the p-value for C.Liberibacter WDYF = 0.00716 will be 0.0072 if using 4 decimal places, and 0.007 if using three decimal places (which is normal for most journals).
Fig. S7 LDA score image if very nice as simplifies distribution across sampled fruits.

Author Response
Dear Reviewer,
Thank you for your comments. We have carefully revised the manuscript according to the review comments. We have uploaded the manuscript and supplementary file. Please see the attachment for revision.
Looking forward to your good reply.
Thank you.
Yours sincerely,
Feiyan Wang

Reviewer 2 Report
The manuscript titled "Diversity Analysis and Function Prediction of Bacterial Communities in the Different Colored Pericarp of Citrus reticulata cv. ‘Shatangju’ due to Candidatus Liberibacter asiaticus infection" aims to examine the impact of Huanglongbing (HLB) on the variety of bacteria present, utilizing microbiota analysis based on 16S ribosomal DNA (rDNA). Additionally, the study aims to delve into the anticipated roles and functions of the core microorganisms within HLB-infected citrus pericarps. The study reveals significant dissimilarities in the diversity and makeup of microbial communities between normal and abnormal pigment pericarp tissues of ripe fruits infected by Huanglongbing (HLB). The impact of HLB with varying concentrations on the relative abundance of the microbial population was found to be distinct. This study provides novel insights into comprehending the composition of the microbiome enriched in HLB-affected citrus pericarps and its influence on plant health. While the topic is of significant relevance and general interest to the journal's readership, several concerns need to be addressed before publication.
· The authors are highly recommended to avoid using a personal pronoun (e.g., We, our, etc.); they can use the third party in the past tense's passive voice.
· The authors are strongly advised to carefully review the manuscript to address grammar and other editing issues.
· To ensure reader comprehension, it is essential to provide the full name associated with any abbreviation at its first mention in the manuscript. This practice enables readers who may not be familiar with the abbreviated terminology to follow along and understand the content. For example, COGs in line 30. Also, provide full names for any abbreviation in the figures caption.
· In the Material and Methods section, it is important to include or complete the sources of all chemicals, software, and equipment by adding the city, state, and country information. This additional detail provides readers with specific information about where these items were sourced, ensuring transparency, and facilitating reproducibility.
· In line 114, add respectively.
· Figures 2, 3, 4, 5, and 8 need to be provided in a high resolution and bigger size. The current format is very hard to follow.
Minor editing of English language required
Author Response
Dear Reviewer,
Thank you for your comments. We have carefully revised the manuscript according to the review comments. We have uploaded the manuscript and supplementary file. Please see the attachment for the revision.
Looking forward to your good reply.
Thank you.
Yours sincerely,
Feiyan Wang

Reviewer 3 Report
In general is an interesting work, but I have some comments:
Abstract:
- Genus of bacteria goes in italic
- Don't use abbreviations not fully described in the abstract r any other place in the text
- Not necessary to use a letter code for the functions in the abstract. Do they respond to any criteria?
Introduction:
- Main text need to be justified in the frames, as editing of the journal requires
- Not sure, but Candidatus Liberobacter should be in italic
- Again genus for strains go in italic
- In general is good, but maybe too short
Results:
- The way to indicate panel letters is not the required by the journal
- The abbreviations are defined later, but this is the first time you use in the main text (figures, result description...), so please use the full naming before to use the abbreviations
- 'P value' is not written in capital letters
- The multi-lines to express significance in the figures may be confusing, consider to use letter to same level of significance in data
- Figure 2: Negative values are not possible in diversity indexes, so try to crop the graph to only positive values
- Percentages are representative, but the error for them was calculated?
- Page 4 is full of precious information, however the amount of information is making it very confusing. It's better to filter the most relevant results or those that really deserve to be mentioned or are significant for the understanding of the process. Alternatively, leave all the info, but find some other way to express it. This stile make it very repetitive and difficult to extract the relevant information
- Probably not necessary to express all the p.values, just mention they are signification and report the minimal p.value the significance was set up
- Some orthographical mistakes
- The correlation analysis is not string for me: the assumptions are barely justified and vaguely described, so I cannot conclude them as solid assumptions. It's valuable, but the expression in here is not fully correct
- norank_f_Mitochondria is not a bacterium....
- The bacterial functions seem like a machine-made report... if the function is 'unknown', is not logical to express here as the same relevance as other data....
- 'Metabolism' was the most abundant KEGG... Seriously? This is meaningless, it's the most logical result.... Remember this is not a report, this is a scientific publication...
Discussion:
- Nice, but some assumptions are not strongly back enough to propose here, specially with correlations, bacteria functions and metabolites. You need more evidences
Mostly ok, need to check some minor orthographical mistakes
Author Response

(The authors gave the same response as above.)

Round 2
Reviewer 1 Report
Authors corrections noted from previous reviews, however a few minor corrections still needed see Lines 22 in abstract use the abbreviation CLas. --do not need to spell out here so use the abbreviation.
Text Lines 65, reword as: '...endosphere in leaves or rhizosphere in roots'.
line 70 -71. reword as: '...pigmentation of the fruit exocarp (the outer peel), but can also affect the diversity of bacteria in the entire citrus fruit pericarp.'
All abbreviations of CLas need to be checked as several are now Clas 'abstr lines 19, 27, 31, Text: Lines 69, 259, 261,265, 272,280, 291,309, 356, 363
After these corrections I think the manuscript will be ready to publish.

Author Response
Dear Reviewer,
Thank you for your comments. We have carefully revised the manuscript according to the review comments. We have uploaded the manuscript and supplementary file. Please see the attachment.
Looking forward to your good reply.
Thank you.
Yours sincerely,
Feiyan Wang

Reviewer 2 Report
The authors responded to all of my comments, and the manuscript can be published unless there are other concerns with the editor or other reviewers.
Miner editing is required!
Author Response
Dear Reviewer,
Thank you for your valuable comments. We have uploaded the revised manuscript and supplementary file.
Looking forward to your good reply.
Thank you.
Yours sincerely,
Feiyan Wang
Reviewer 3 Report
The quality seems better after corrections. Thanks for taking them so carefully. I still think this requires extra test, but now is more moderated in assumptions. I would recommend reinforcing the idea of this work as a bioinformatic approach (in silico) to avoid other circumstances, in case no extra test are considered. Leave the final decision to the editor then.
Author Response

(The authors gave the same response as above.)
